# ADAPTIVE TEXT TRANSFORMATIONS DEFEND AGAINST DATASET INFERENCE ATTACKS

## ABSTRACT

There has been increasing legal interest in identifying possible copyright viola-
tions committed by Large Language Model (LLM) trainers. Many works devel-
oping individual membership inference attacks have recently been shown to be
weaker than previously thought, due to implicit distributional shifts. To combat
this, progress has been made by considering large datasets and aggregating mul-
tiple different membership attacks. A hidden assumption in these methods is that
if the LLM has improperly used a dataset, the LLM was trained on that exact
dataset. By challenging this assumption, we demonstrate, to our knowledge, the
first failure of any large-scale dataset inference (DI) attack. In particular, we study
LLM fine-tuning for both short and long text datasets. We adaptively transform
the datasets before fine-tuning, enabling an increase in model performance (com-
pared to a base model not trained on the dataset) while avoiding dataset inference.
In the case of long texts, we find that text summarization followed by rephrasing
substantially reduces the success probability of DI in our setting from over 95% to
less than 5%. We also develop a new theoretical formulation of dataset inference
specifically tailored to LLMs, which explains the effectiveness of our method and
sheds light on how parameters, such as the number of training epochs, can affect
dataset inference.

## 1    INTRODUCTION

Large language models are trained on internet-scale corpora, and there is an increasing interest in
identifying copyright violations that occur within this large space. For example, the recent lawsuit
between The New York Times and OpenAI/Microsoft (Grynbaum & Mac, 2023) highlights the risks
associated with training on copyrighted data. Anthropic's massive $1.5 billion settlement on 500,000
copyrighted books (Metz, 2025) shows that these risks can be realized and lead to massive financial
penalties for LLM trainers. The corresponding machine learning problem is to determine, given an
LLM and a dataset, whether the LLM was trained on the dataset.

Historically, techniques focusing on understanding the training data of LLMs have primarily focused
on membership inference, in which a single document is assessed for membership in the training set
of an LLM (Shokri et al., 2017). However, it has been shown that existing MIA methods may have
incorrect success estimates due to distributional shifts. In particular, test data (i.e., guaranteed to be
private) often comes after the training date cutoff of the LLM to ensure there is no leakage (Maini
et al., 2024), but this causes the splits to be separated in time, so an LLM can distinguish between
them even if it was not trained on the hypothesized member.

Recent work in this area has focused on dataset inference (Maini et al., 2024; Puerto et al., 2025),
where we assess collections of documents for existence in training data rather than a single element,
with the hope that weak predictions can be combined into a strong one.

To our knowledge, due to their experimental setup, current LLM dataset inference methods assume
that if training occurred, it occurred on exactly the copyrighted dataset hypothesized to have been
stolen. For example, (Maini et al., 2024) studies the performance of dataset inference on splits of
the Pile, on which Pythia models were known to have been trained.

Effectively, this does not account for adversarial behavior by the LLM trainer. We find that even simple countermeasures can substantially degrade the performance of dataset inference. More specifically,

1. We provide a technique that substantially decreases the likelihood of true positive detection of the (Maini et al., 2024) attack while maintaining performance in false positive cases.

2. We demonstrate that it is possible to evade dataset inference through dataset transformation while maintaining benchmark performance.

3. We provide a theoretical formulation and analysis of dataset inference that significantly expands upon prior work (Maini et al., 2021) and offers a general functional form for the dataset inference success rate.

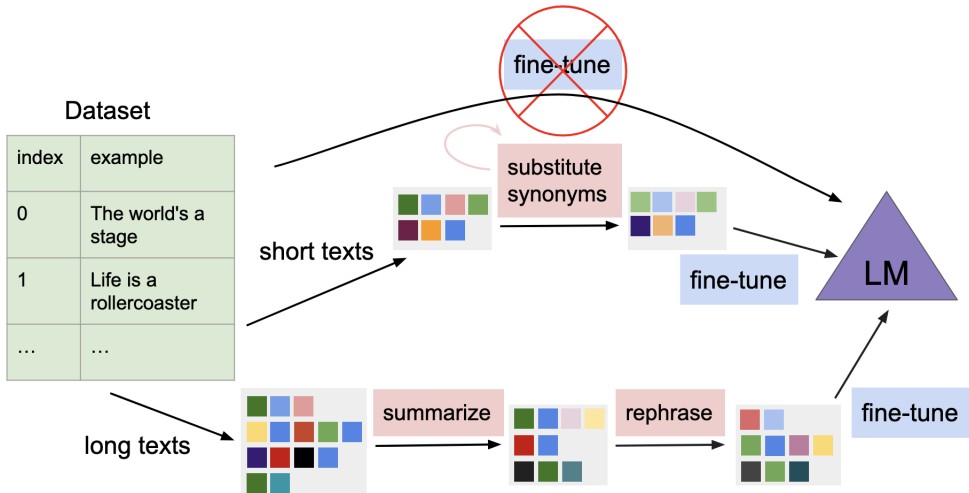

Figure 1: We adaptively transform datasets with per-document changes before fine-tuning, blocking most dataset inference attacks. For short texts, we use synonym substitution repeatedly until the embedding similarity with the original is under a specified threshold. For long texts, we summarize the text and then rephrase it. In either case, we preserve the overall meaning of a text, and instead of fine-tuning on the original, we fine-tune on the transformed text. For evaluation of short text transformations, we successfully (while evading DI) fine-tune GPT-2 on multiple benchmarks (GSM-MC, BoolQ, MNLI) and achieve over 90% of the total possible benchmark improvement. For long text transformation evaluation, we fine-tune multiple sizes of Pythia models on 12 datasets from The Pile, and can evade dataset inference in over 95% of cases. In contrast, without our method, dataset inference succeeds in over 95% of cases.

## 2 RELATED WORK

**Membership Inference Attacks**  Membership inference is a smaller-scale version of dataset inference in which we are interested in only a single token sequence, such as a sentence. Membership inference attacks have been well studied in several prior works, with numerous proposals for attack methods (Shokri et al., 2017; Mattern et al., 2023; Shi et al., 2024). There have been multiple evaluation methods for these attacks as well (Fu et al., 2024; Liang & You, 2024). Zhang et al. (2025) shows that membership inference attacks can have unbounded false positive rates. Related problems with applications in model inversion (Fredrikson et al., 2015) and stealing training data (Carlini et al., 2021) have also received attention.

**Dataset Inference**  Maini et al. (2024) showed that most membership inference attacks are roughly no better than random chance. In particular, the positive results shown in prior works occur due to a confounding variable of distributional shifts over time. A key part of the new problem setting in Maini et al. (2024) is having two separate variations of a dataset, both drawn from the same distribution (a "dataset space") in which only one could have been used for LLM training, for example,

with several versions of a book chapter, only one of which is ultimately published. Aggregating over multiple membership inference attacks is also crucial for amplifying any weak signal that a single attack may reveal. On the theoretical side, Maini et al. (2021) contains an important analysis of dataset inference, which we compare to extensively.

# 3 THEORETICAL JUSTIFICATION

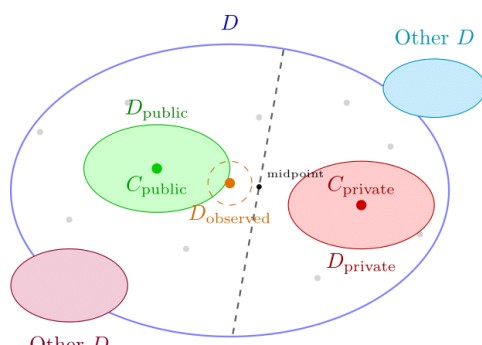

Figure 2: Theoretical formulation of LLM Dataset Inference. In our model of dataset inference, there is a dataset $D$, two independent and identically distributed (iid) samples $D_{private}$ and $D_{public}$, and we can compute the corresponding sample centers, $C_{public}$ and $C_{private}$. We are able to noisily evaluate the average data point, $D_{observed}$, of the nearby training distribution. The decision boundary between public and private splits is the perpendicular bisector hyperplane between the centers, and we check the side $D_{observed}$ is on.

## 3.1 THEORETICAL GOALS

A popular theoretical justification of dataset inference is given by Maini et al. (2021). We identify several shortcomings of the prior analysis and thus corresponding goals for a new formulation specific to LLMs:

1. The analysis of Maini et al. (2021) relies on the victim model being a linear classifier, with an exact pre-specified learning algorithm and a constant learning rate. The authors justify this by noting that more parameters lead to even greater success of dataset inference. While this may be true in practice due to memorization (Zhang et al., 2017; Feldman, 2020; Carlini et al., 2019), it does not account for the randomness present in the training or generation process of an LLM or, significantly, our paper's method.

2. Both LLM trainers and copyright holders will be interested in knowing how the specific dataset and training dynamics affect dataset inference success probabilities. The analysis in Maini et al. (2021) does not reflect the influence of training parameters such as learning rate, number of epochs, number of parameters in the model, or any parameters of the particular dataset.

3. The conclusion of Maini et al. (2021) is that the number of training samples (i.e., dataset size) does not influence the probability of dataset inference succeeding. This result is inconsistent with recent studies, as well as the paper's own results (Puerto et al., 2025; Maini et al., 2024; 2021), which show that the success probability increases rapidly as the dataset size increases. A theoretical justification of this phenomenon would help contextualize prior results and inform possible future work.

4. The analysis in Maini et al. (2021) only considers the leading order terms in $N$, the dimensions of a relevant space. The effects of the lower-order terms are unclear, and the result is a conclusion that holds in expectation. We aim to more precisely characterize the success probability and provide a high-probability bound.

To our knowledge, a formulation and corresponding theoretical result that achieves even one of these goals (e.g., explaining theoretically why LLM dataset inference becomes more successful as the dataset size increases) would be novel. We believe that we adequately address all of them.

## 3.2 THEORETICAL FORMULATION

We view all the training data coming to the LLM as existing in a latent space.

Let $R^n$ be the the latent space. The dataset distribution is assumed to be an isotropic Gaussian:

$$D \sim \mathcal{N}(\mu, I_N).$$

We draw $n$ samples to create two dataset splits (public and private), and have access to the dataset centers. This setup is inspired by Maini et al. (2024), which assumes access to two IID splits coming from a larger distribution.

The dataset centers can be interpreted as independent Gaussian perturbations of $\mu$:

$$C_{\text{pub}} = \mu + A, \qquad C_{\text{priv}} = \mu + B,$$

with

$$A,\ B \overset{\text{iid}}{\sim} \mathcal{N}\big(0,\ \sigma_{\text{split}}^2 I_N\big).$$

A dataset inference attack is interpreted as the ability to observe the average over the training data of an LLM in the region close to the dataset, e.g., by using a prompt that elicits an appropriate conditional probability distribution.

In particular, we observe a point $D_{\text{obs}}$ which is a noisy weighted interpolation between the population mean and one of the centers. If the true source is "pub" then

$$D_{\text{obs}} = \epsilon C_{\text{pub}} + (1 - \epsilon)\mu + \eta = \mu + \epsilon A + \eta,$$

and similarly if the source is "priv" then $D_{\text{obs}} = \mu + \epsilon B + \eta$. Here

$$\eta \sim \mathcal{N}\big(0,\ \sigma_{\text{obs}}^2 I_N\big)$$

is observation noise, and $0 \leq \epsilon \leq 1$ is the mixture weight assigned to the split of $D$.

The observation noise accounts for the randomness that can occur during the attack, as well as methods that can be used to noise the dataset, such as ours.

The mixture weight accounts for the unrelated training data close in the latent space to the splits under consideration. The unrelated training data has weight $1 - \epsilon$. Note that if there were no consideration of mixture, $D_{obs}$ would simply be a noisy observation of one of the dataset centers. However, this does not make sense, particularly in the case of small $m$.

The decision rule is:

$$\text{decide "pub" if and only if } \|D_{\text{obs}} - C_{\text{pub}}\| < \|D_{\text{obs}} - C_{\text{priv}}\|.$$

This rule can be interpreted as finding the perpendicular bisector hyperplane between $C_{public}$ and $C_{private}$ and checking which side $D_{observed}$ is on. Due to our formulation of $D_{observed}$ being closer in expectation to the dataset center it is trained on, this decision rule is the best possible, assuming equal priors.

Formally, the success probability of dataset inference under our model is the probability that the above decision rule returns the correct class.

### 3.3 THEORETICAL RESULTS

We now provide theoretical results for the setting described in the previous section.

**Theorem 1.** *Suppose we denote $p_{success}$ as the success probability of dataset inference. Then, there exists a constant $C$ such that with probability $1 - \delta$, the following holds:*

$$\Phi\left(\frac{2\epsilon N\sigma_{split}^2 - 4\sigma_{split}^2\sqrt{Nt}}{2\sigma_{obs}\sqrt{2N\sigma_{split}^2 + 8\sigma_{split}^2\sqrt{Nt}}}\right) \leq p_{success} \leq \Phi\left(\frac{2\epsilon N\sigma_{split}^2 + 2\sigma_{split}^2\sqrt{Nt}}{2\sigma_{obs}\sqrt{2N\sigma_{split}^2 - 8\sigma_{split}^2\sqrt{Nt}}}\right)$$

*where $t = \frac{\log(6/\delta)}{C}$ and $\Phi$ is the Gaussian CDF.*

*Proof.* See Appendix A. □

**Corollary 1.** *To leading order in $N$, the success probability of LLM dataset inference is*

$$p_{success} \approx \Phi\left(\frac{\epsilon\sigma}{\sigma_{obs}} \cdot \sqrt{\frac{N}{2m}}\right)$$

*where $m$ is the dataset size and $\sigma$ is the dataset variance.*

*Proof.* See Appendix B. □

### 3.4 INTERPRETATION

In our model, $\epsilon$ is regarded as a weighting parameter for the dataset being trained compared to other nearby distributions in the latent space.

In the regime of small $m$, $\epsilon$ will increase linearly in $m$ because the dataset as a whole has correspondingly more relative weight. For a similar reason, $\epsilon$ will scale with the learning rate $\alpha$ and the number of epochs, $E$. We additionally propose that the dimensionality of the appropriate latent space increases linearly with the number of parameters in the model being tested, $\theta$.

Along with Corollary 1, we have that in the small-$\epsilon$ regime, we have

$$p_{success} \approx \Phi\left(\frac{m\alpha E\epsilon_0\sigma}{\sigma_{\text{obs}}} \cdot \sqrt{\frac{N_0\theta}{2m}}\right) = \Phi\left(\frac{\alpha\sigma E\epsilon_0}{\sigma_{\text{obs}}} \cdot \sqrt{\frac{N_0\theta m}{2}}\right) \tag{1}$$

Similarly, in the large-$\epsilon$ regime (close to 1), we have

$$p_{success} \approx \Phi\left(\frac{\sigma}{\sigma_{\text{obs}}} \cdot \sqrt{\frac{N_0\theta}{2m}}\right) \tag{2}$$

We can justify assuming that $N$ is large in Corollary 1 because the latent token space of GPT-2 is already as much as 1600-dimensional (Radford et al., 2019), and we are dealing with datasets for which each element is many tokens, not just one.

In the case of small $\epsilon$, Equation 1 shows that success probability increases with $m$. This relationship explains the results from multiple works on dataset inference, which have shown that in practice the attack metrics (e.g., AUROC, probability of detection) improve as the sample size increases (Maini et al., 2021; 2024; Puerto et al., 2025).

In the case of large $\epsilon$, Equation 2 shows that the success probability decreases with $m$. This decrease has a corollary which, at first glance, seems absurd: an LLM trainer who steals and trains on an extremely large amount of copyrighted data is less likely to be detected (at least through model-based methods) than a trainer who steals a moderate amount of data. However, we argue that this is actually intuitive. In particular, as the sample size increases, the centers of the private and public splits will become close together in latent space (the typical distance between the corresponding centers is of order $\sigma/\sqrt{m}$, which goes to 0 as $m$ increases). The other datasets, which may be nearby, become irrelevant. However, if a method with a constant per-element level of noise, such as ours (or even a traditional stochastic training algorithm), is used, the tiny separation between centers becomes irrelevant, and the DI attack turns into a coin flip.

Finally, $\sigma$ can be regarded as a property of the dataset distribution, where a dataset that has many very diverse examples has a higher value. We can interpret $\sigma_{obs}$ as a constant representing the strength of a dataset inference attack in comparison with defense methods, with the sharpest attacks having lower values and the most effective defenses having higher values.

## 4 METHOD

We expect synonym substitution to be effective because it adds noise to the label parameters; in other words, there can be crossover between the public and private splits. We expect it to still retain much of the original meaning due to the close semantic similarity of synonyms (Taraba, 2020; Bhagat &

---

**Algorithm 1** Evading Dataset Inference on Short Benchmarks

---

**Require:** A pre-trained LLM $L$, a possibly copyrighted dataset $D$ with private version $D_{private}$, an embedding model $emb()$, a synonym substituter $sym$, a fine-tuning method fine-tune(), a dataset inference method attack(), and a list of similarities to consider, $similarities$.
1:   $i = 0$
2: **while** $i < \text{len}(similarities)$ **do**
3:    $D' = []$
4:    **for** $d \in D$ **do**
5:     $e_{init} = emb(d)$
6:     **while** cosine-similarity$(e_{init}, emb(d)) > similarities[i]$ **do**
7:      $d \leftarrow sym(d)$
8:     **end while**
9:     Add $d$ to $D'$.
10:    **end for**
11:    $L' \leftarrow \text{fine-tune}(L, D')$
12:    $p = \text{attack}(L, D, D_{private})$
13:    **if** $p > 0.1$ **then**
14:     **return** $L'$
15:    **end if**
16:    $i \leftarrow i + 1$
17: **end while**
18: **return** $L$

---

**Algorithm 2** Evading Dataset Inference on Large Corpora

---

**Require:** A dataset $d$, a pre-trained LLM $L$, and a fine-tuning method fine-tune().
1:   $d' = []$
2: **for** text $t$ in $d$ **do**
3:    $t' \leftarrow$ Call Mistral-7B-Instruct-v0.3 on prompt
     "Summarize the following \n Text: $\{t\}$\nTLDR: "
4:    $t' \leftarrow$ Call Mistral-7B-Instruct-v0.3 on prompt
     "Rephrase the following text with synonyms and rearrangement.
     Do not output anything but the rephrased text.\n\n Text: $\{t'\}$ \n\n Rephrased: "
5:    Append $t'$ to $d'$
6: **end for**
7: $L' \leftarrow \text{fine-tune}(L, D')$
8: **return** $L'$

---

Hovy, 2013). For example, the argument to $\Phi$ in Section 3 may change from 2 to 1 due to $\sigma_{obs}$ doubling. In this case, the success probability goes from $0.977$ to $0.841$, a substantial reduction.

We expect summarization to be effective because it is a form of dimensionality reduction (Rodrigues et al., 2025; Bartakke et al., 2021; Abdelrahman et al., 2023). We empirically observe that our method reduces the number of tokens by approximately a factor of 3 (see Appendix D). Therefore, we estimate that $N_0$ from Section 3 goes down by a factor of 3. Since we also rephrase, this increases $\sigma_{obs}$ as in the synonym substitution case. If we suppose $\sigma_{obs}$ goes from 1 to $\sqrt{3}$, it could be that the argument of $\Phi$ goes from 3 to $3 \cdot \frac{1}{\sqrt{3}} \cdot \frac{1}{\sqrt{3}} = 1$. These values imply a naive DI success probability of $0.999$, whereas our method yields a success probability of $0.841$.

Note that in both cases, a final probability of $0.841$ may seem high. However, some notion of statistical significance is involved because, in a real legal case, care would be taken to avoid judgments based on a statistic that could have occurred randomly. At a $p = 0.1$ level, a probability of $0.841 < 0.9$ is not significant.

# 5 RESULTS

Due to constrained computation resources, we analyze the fine-tuning setting. However, we expect that dataset inference attacks are more successful in this setting (Puerto et al., 2025) due to the lower learning rate used during pre-training (Team, 2025), as well as the internet-scale corpus employed. In our setting, a true positive is when a restricted dataset was used for training and this is detected. A false positive is when a restricted dataset was not used for training, but the test declared that is was used.

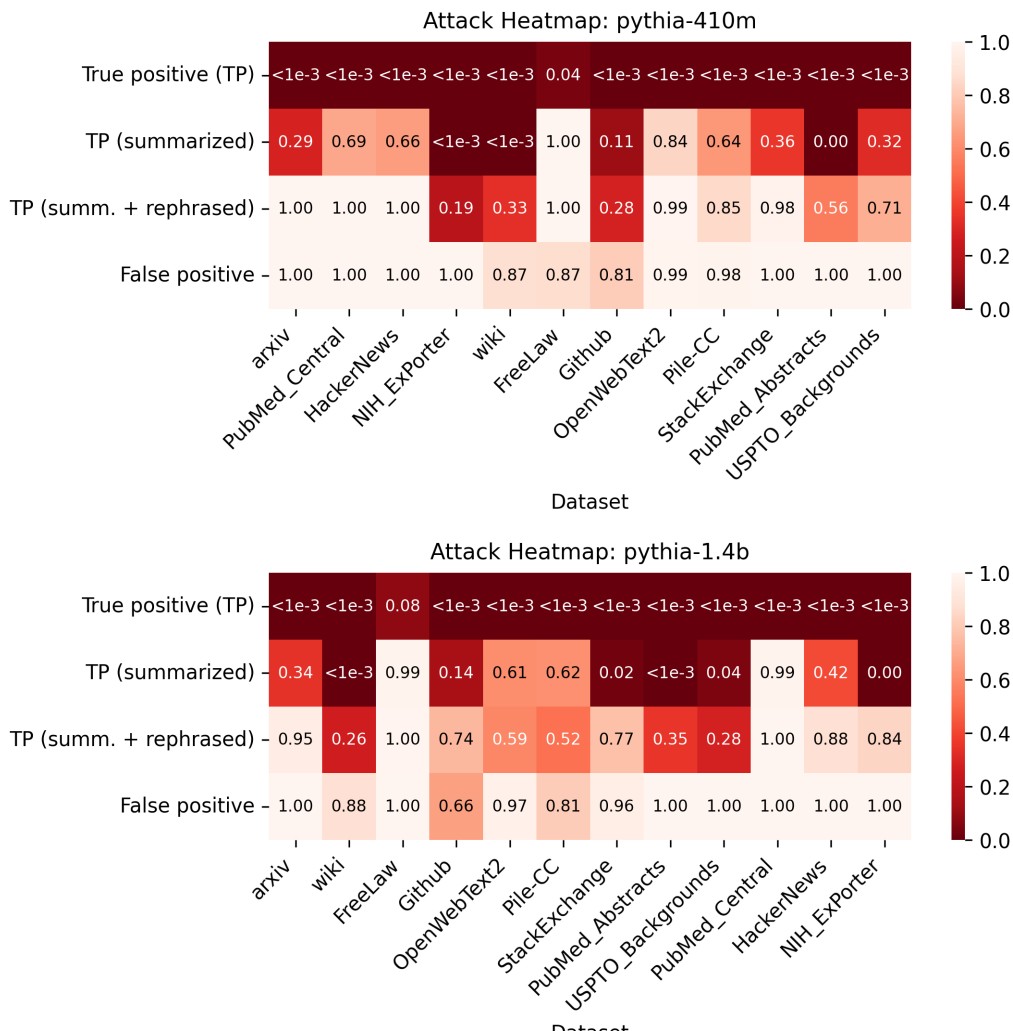

Figure 3: Adaptively transforming Pile data with summaries and rephrasings. Each number in the heatmap represents an aggregate $p$-value from Equation 3. We want the $p$-value to be as high as possible because it indicates a low likelihood of a DI attack succeeding. Almost uniformly across datasets, the $p$-value for the true positive is very low while the false positive is high. These results are what we expect and are analogous to Figure 4 in Maini et al. (2024). Our method, which involves summarization followed by rephrasing, achieves relatively high p-values across all datasets for both Pythia-410m and Pythia-1.4b. We use a threshold of $p = 0.1$, as in Maini et al. (2024). The high p-values demonstrate an ability to evade dataset inference, because DI is expected to give low $p$-values (chance of training on the private split) in the true positive case.

Table 1: Number of Pile subsets evading LLM Dataset Inference.

| Method | Pythia-70m | Pythia-410m | Pythia-1b | Pythia-1.4b |
|---|---|---|---|---|
| Baseline | 1 | 0 | 0 | 0 |
| Summarized | **11** | 8 | 6 | 7 |
| Summarized + Rephrased | **11** | **12** | **12** | **12** |

### 5.1 SETUP FOR PILE EXPERIMENTS

For each dataset in The Pile, we train on the first 1000 validation examples. This method requires us to restrict our attention to datasets within The Pile that have at least 1000 validation and test examples. We include all such datasets, except for DM Mathematics, which consists of each element as a long list of math questions, making it unsuitable for summarization. This elimination leads to a total of 12 datasets.

After training on each of these, we run the attack of (Maini et al., 2024), using the 1000 validation examples as the possibly copyright-violated split and the first 1000 test examples as the private split.

We aggregate probabilities in the same way as (Maini et al., 2024), with the formula

$$P_{combined} = 1 - \exp\left(\sum_{i=1}^{n} \log(1 - p_i)\right) \tag{3}$$

We run five separate attacks for each fine-tuned LLM on each dataset. We train on multiple sizes of LLMs, so we adaptively adjust the learning rate according to the formula $\alpha = \frac{1.5}{\theta}$, where $\theta$ is the number of model parameters.

The results are pictured in Figure 3 and Table 1. Overall, we succeed in defending against the dataset inference attack in most cases, in particular for the summarize + rephrase technique, it defends the LLM in $\frac{23}{24} \approx 96\%$ of cases.

### 5.2 SETUP FOR BENCHMARK EXPERIMENTS

In addition to the usual participants, such as an LLM and a dataset, Algorithm 1 also requires an embedding model and a synonym augmenter. We use all-MiniLM-L6-v2 as the embedding model, and nlpaug as the synonym augmenter.

For each benchmark, we fine-tune the model on training examples with a benchmark-specific format and then evaluate it on the validation split. We focus on multiple-choice benchmarks so they are easy to evaluate. For the DI attack, we employ the method presented in (Maini et al., 2024), running each attack twice and averaging the p-values.

The results are pictured in Figure 4. Note that we can fine-tune safely (without detection) in all three cases and achieve significant benchmark performance improvements.

### 5.3 ABLATION STUDY

In the case of short texts, the results for higher similarities than the green dot can be regarded as a form of ablation (i.e., not running the method to completion). In most cases, the evasion probability is very low, and all of the iterations are substantially needed to evade DI.

In the case of long texts, we provide results on the effect of just summarizing in Figure 3. We see that in many cases, dataset inference would succeed without the rephrasing step. In particular, in 8 out of 24 cases across Pythia-70m and Pythia-410m, dataset inference succeeds, which is significantly higher than the corresponding 1 out of 24 cases if rephrasing also occurs.

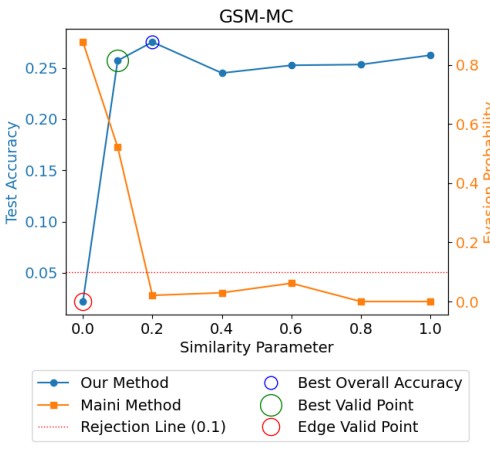

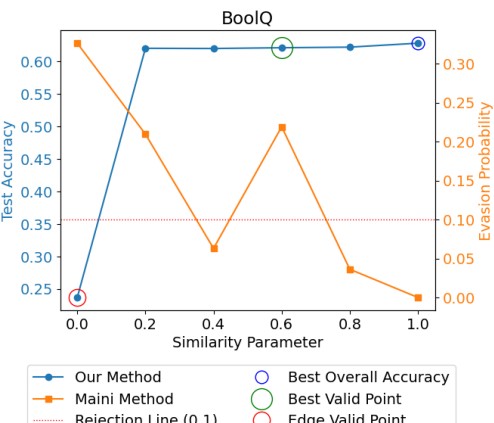

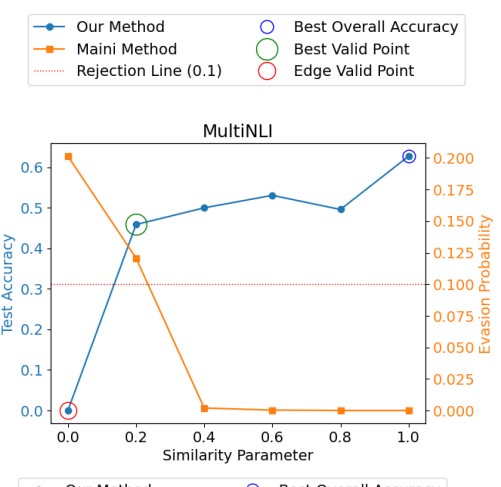

Figure 4: Experiment fine-tuning GPT-2 on benchmarks. The orange dots represent the probability of the alternative hypothesis using the (Maini et al., 2024) method. The blue dots represent performance on the respective benchmark test sets. The red circle marks the point with the highest accuracy such that the evasion probability is at least 0.1 and the similarity is either 0 or 1, if such a point exists. The green circle represents the point with the highest accuracy, ensuring that the evasion probability is at least 0.1, regardless of the similarity parameter. The blue circle represents the highest possible accuracy across all similarity parameters.

## 6 DISCUSSION

Our method is largely successful across all tested Pile datasets, but it can be observed that some datasets are more successful than others. For example, Wiki and PubMed Abstracts both have low $p$-values across all model sizes for the summarize method, and this is also observed to some extent for the summarize + rephrase method. To apply our method in practice, the parameters of the method should vary depending on the properties of the specific dataset. We did not address this to avoid complications from over-parameterization and overfitting.

Recall that we adaptively varied the learned rate as $\alpha = 1.5/\sqrt{\theta}$. Even though the number of parameters of large closed-source LLMs is often not released (OpenAI, 2024), adaptive variation is a feasible practice because the learning rate is selected by the LLM trainer, not the attacker. Recall that our theoretical formulation of LLM dataset inference predicts that success probability in cases of small training set size should be approximately $\Phi\left(\frac{\alpha\sigma E\epsilon_0}{\sigma_{\text{obs}}} \cdot \sqrt{\frac{N_0\theta m}{2}}\right)$.

The linear dependence inside the $\Phi$ on $\alpha$ and $E$ appears to be well-motivated by the theoretical formulation. However, the dependence on the number of parameters, $\theta$, is less clear (in our formula, we suppose that there is a $\sqrt{\theta}$ dependence inside the $\Phi$). If we plug in our chosen learning rate value we have $\Phi\left(\frac{\alpha\sigma E\epsilon_0}{\sigma_{\text{obs}}} \cdot \sqrt{\frac{N_0\theta m}{2}}\right) \Phi\left(\frac{1.5\sigma E\epsilon_0}{\sigma_{\text{obs}}} \cdot \sqrt{\frac{N_0 m}{2}}\right)$. In other words, we expect the $\alpha$ and $\sqrt{\theta}$ terms to cancel out due to our choice of learning rate, resulting in similar results across models. Indeed, the heatmaps across models are qualitatively similar (see Figure 3 and Figure 5), as shown in Table 1 as well. This similarity provides initial evidence for the $\sqrt{\theta}$ dependence, but additional research is needed to validate or refute this finding. Another limitation is the strict assumptions on dataset properties, and experiments on real-world datasets are needed in the future.

## 7 CONCLUSION

The strong results in the past literature on dataset inference (Maini et al., 2024; Puerto et al., 2025) can be viewed as relying on the assumption that we have access to the exact dataset on which an LLM was trained. This assumption can be easily broken in practice by the LLM trainer, specifically by taking a copyrighted dataset and transforming the elements in some way before training on them. Existing techniques are not robust to this method and actually fail even in simple semantic-preserving scenarios.

These results underscore a critical limitation in current dataset inference methodology. Future work should explore principled methods that remain effective under text transformations, and further investigate trade-offs between utility preservation, attack resistance, and legal or ethical guarantees.

Our theoretical results indicate the existence of a "tipping point", where, after a certain number of samples, dataset inference starts becoming harder, contrasting with the small sample behavior of becoming easier (i.e., higher success probabilities as samples grow). Additional research is needed to determine if such a point exists and, if so, to identify relevant practical bounds. Even the existence of such a point may foreshadow a need to eventually move beyond the traditional assumptions of dataset inference.

## 8 ETHICS STATEMENT

Our adaptive text transformation method can be viewed as a mechanism for training on copyrighted datasets while avoiding adverse legal consequences. We aim to expose the potential issues with traditional approaches to dataset inference, enabling more research in this area. We believe this will allow for more robust copyright protections in the future. Additionally, our method can be utilized positively to prevent the leakage of training data. For example, if it is known that an LLM is trained on some arXiv collections but not others, a dataset inference attack could plausibly be used to extract which ones it was trained on. This information could be regarded as proprietary by the model creator, necessitating protection.

## 9 REPRODUCIBILITY STATEMENT

The mathematical claims made in the paper are proven in the main text or the appendix. All code used to generate results and figures will be made public upon acceptance, along with documentation on how to reproduce the results.

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
