## A    PROOF OF THEOREM 1

By symmetry, it suffices to compute the success probability conditional on the true class being "pub".

### A.1    CONDITIONAL DISTRIBUTION OF SUCCESS PROBABILITY ON A AND B

Assume the true class is "pub", so $D_{\text{obs}} = \mu + \epsilon A + \eta$. Compute the two residuals:

$$D_{\text{obs}} - C_{\text{pub}} = (\mu + \epsilon A + \eta) - (\mu + A) = -(1-\epsilon)A + \eta,$$
$$D_{\text{obs}} - C_{\text{priv}} = (\mu + \epsilon A + \eta) - (\mu + B) = \epsilon A - B + \eta.$$

Define the test statistic

$$\Delta := \|D_{\text{obs}} - C_{\text{pub}}\|^2 - \|D_{\text{obs}} - C_{\text{priv}}\|^2.$$

We have

$$\|D_{\text{obs}} - C_{\text{pub}}\|^2 = \| -(1-\epsilon)A + \eta \|^2$$
$$= (-(1-\epsilon)A + \eta) \cdot (-(1-\epsilon)A + \eta)$$
$$= (1-\epsilon)^2 \|A\|^2 - 2(1-\epsilon)A \cdot \eta + \|\eta\|^2.$$

In addition,

$$\|D_{\text{obs}} - C_{\text{priv}}\|^2 = \|\epsilon A - B + \eta\|^2$$
$$= (\epsilon A - B + \eta) \cdot (\epsilon A - B + \eta)$$
$$= \epsilon^2 \|A\|^2 + \|B\|^2 + \|\eta\|^2$$
$$- 2\epsilon A \cdot B + 2\epsilon A \cdot \eta - 2B \cdot \eta.$$

Substituting the expansions,

$$\Delta = \left((1-\epsilon)^2\|A\|^2 - 2(1-\epsilon)A \cdot \eta + \|\eta\|^2\right)$$

$$- \left(\epsilon^2\|A\|^2 + \|B\|^2 + \|\eta\|^2 - 2\epsilon A \cdot B + 2\epsilon A \cdot \eta - 2B \cdot \eta\right).$$

The $\|\eta\|^2$ terms cancel, leaving

$$\Delta = \left((1-\epsilon)^2 - \epsilon^2\right)\|A\|^2 - 2(1-\epsilon)A \cdot \eta - \|B\|^2$$

$$+ 2\epsilon A \cdot B - 2\epsilon A \cdot \eta + 2B \cdot \eta.$$

Simplifying the coefficient of $\|A\|^2$, we have

$$(1-\epsilon)^2 - \epsilon^2 = (1 - 2\epsilon + \epsilon^2) - \epsilon^2 = 1 - 2\epsilon.$$

Grouping the $\eta$-dependent terms, we have

$$-2(1-\epsilon)A \cdot \eta - 2\epsilon A \cdot \eta + 2B \cdot \eta = -2A \cdot \eta + 2B \cdot \eta = -2(A - B) \cdot \eta.$$

$$\Delta = -2(A - B) \cdot \eta + (1 - 2\epsilon)\|A\|^2 + 2\epsilon A \cdot B - \|B\|^2. \tag{4}$$

We expand each squared norm explicitly. Recall

$$D_{\text{obs}} - C_{\text{pub}} = -(1-\epsilon)A + \eta, \qquad D_{\text{obs}} - C_{\text{priv}} = \epsilon A - B + \eta.$$

The decision is correct if and only if $\Delta < 0$. Conditional on $A, B$ the only randomness in $\Delta$ is through $\eta$. Since $\eta \sim \mathcal{N}(0, \sigma_{\text{obs}}^2 I_N)$ and $A, B$ are fixed in this conditioning,

$$(A - B) \cdot \eta \sim \mathcal{N}\left(0, \ \sigma_{\text{obs}}^2\|A - B\|^2\right).$$

Therefore, conditional on $A, B$,

$$\Delta \sim \mathcal{N}\left(\mu_\Delta(A, B), \ 4\sigma_{\text{obs}}^2\|A - B\|^2\right).$$

where

$$\mu_\Delta(A, B) = (1 - 2\epsilon)\|A\|^2 + 2\epsilon A \cdot B - \|B\|^2. \tag{5}$$

Hence, the conditional success probability (true = pub) is

$$\Pr\left(\text{correct} \mid A, B\right) = \Pr(\Delta < 0 \mid A, B) = \Phi\left(-\frac{\mu_\Delta(A, B)}{2\sigma_{\text{obs}}\|A - B\|}\right), \tag{6}$$

where $\Phi$ is the standard normal CDF.

The unconditional success probability is the expectation of equation 6 over the Gaussian law of $A, B$:

$$\Pr(\text{correct}) = \mathbb{E}_{A,B}\left[\Phi\left(-\mu_\Delta(A, B)/(2\sigma_{\text{obs}}\|A - B\|)\right)\right]. \tag{7}$$

### A.2 High-dimensional high-probability bound

Fix a failure probability parameter $0 < \delta < 1$.

It is known that as the dimension (i.e., $N$) grows, the 2-norm of a random vector drawn from the normal distribution becomes close to $\sqrt{N}$ and the dot product between two such random vectors becomes close to 0 Vershynin (2018). In particular, we leverage the following high-probability statements Vershynin (2018):

$$\Pr\left\{\left|\|A\|^2 - N\sigma_{\text{split}}^2\right| \geq 2\sigma_{\text{split}}^2\sqrt{Nt}\right\} \leq 2e^{-C_1 t},$$

$$\Pr\big\{|\|B\|^2 - N\sigma_{\text{split}}^2| \geq 2\sigma_{\text{split}}^2\sqrt{Nt}\big\} \leq 2e^{-C_1 t},$$

$$\Pr\big\{|A \cdot B| \geq 2\sigma_{\text{split}}^2\sqrt{Nt}\big\} \leq 2e^{-C_2 t},$$

for some constants $C_1$ and $C_2$.

By a union bound, with probability at least $1 - 6e^{-\min(C_1,C_2)t}$, all three of the following events occur:

$$\Big\{|\|A\|^2 - N\sigma_{\text{split}}^2| \leq 2\sigma_{\text{split}}^2\sqrt{Nt}, \ |\|B\|^2 - N\sigma_{\text{split}}^2| \leq 2\sigma_{\text{split}}^2\sqrt{Nt}, \ |A \cdot B| \leq 2\sigma_{\text{split}}^2\sqrt{Nt}\Big\} \quad (8)$$

Thus to guarantee total failure probability $\leq \delta$ choose

$$t = \frac{\log(6/\delta)}{\min(C_1, C_2)}.$$

From the inequalities in equation 8 we obtain corresponding bounds for $\|A - B\|$. Using

$$\|A - B\|^2 = \|A\|^2 + \|B\|^2 - 2A \cdot B,$$

and applying the three inequalities yields

$$2N\sigma_{\text{split}}^2 - 8\sigma_{\text{split}}^2\sqrt{Nt} \ \leq \ \|A - B\|^2 \ \leq \ 2N\sigma_{\text{split}}^2 + 8\sigma_{\text{split}}^2\sqrt{Nt}. \quad (9)$$

Recall from equation 4

$$\mu_\Delta(A, B) = (1 - 2\epsilon)\|A\|^2 + 2\epsilon \, A \cdot B - \|B\|^2.$$

Using the inequalities in equation 8, we have

$$\mu_\Delta(A, B) \geq (1 - 2\epsilon)\big(N\sigma_{\text{split}}^2 - 2\sigma_{\text{split}}^2\sqrt{Nt}\big) - 2\epsilon(2\sigma_{\text{split}}^2\sqrt{Nt}) - \big(N\sigma_{\text{split}}^2 + 2\sigma_{\text{split}}^2\sqrt{Nt}\big)$$

$$= -2\epsilon N\sigma_{\text{split}}^2 - 2\sigma_{\text{split}}^2\sqrt{Nt},$$

and similarly

$$\mu_\Delta(A, B) \leq (1 - 2\epsilon)\big(N\sigma_{\text{split}}^2 + 2\sigma_{\text{split}}^2\sqrt{Nt}\big) + 2\epsilon(2\sigma_{\text{split}}^2\sqrt{Nt}) - \big(N\sigma_{\text{split}}^2 - 2\sigma_{\text{split}}^2\sqrt{Nt}\big)$$

$$= -2\epsilon N\sigma_{\text{split}}^2 + 4\sigma_{\text{split}}^2\sqrt{Nt}.$$

Thus,

$$-2\epsilon N\sigma_{\text{split}}^2 - 2\sigma_{\text{split}}^2\sqrt{Nt} \ \leq \ \mu_\Delta(A, B) \ \leq \ -2\epsilon N\sigma_{\text{split}}^2 + 4\sigma_{\text{split}}^2\sqrt{Nt}, \quad (10)$$

The denominator in the argument of $\Phi$ is $2\sigma_{\text{obs}}\|A - B\|$.

From equation 9 we obtain,

$$2\sigma_{\text{obs}}\sqrt{2N\sigma_{\text{split}}^2 - 8\sigma_{\text{split}}^2\sqrt{Nt}} \ \leq \ 2\sigma_{\text{obs}}\|A - B\| \ \leq \ 2\sigma_{\text{obs}}\sqrt{2N\sigma_{\text{split}}^2 + 8\sigma_{\text{split}}^2\sqrt{Nt}},$$

We are finally able to state the theorem: with probability at least $1 - \delta$,

$$\Phi\left(\frac{2\epsilon N\sigma_{\text{split}}^2 - 4\sigma_{\text{split}}^2\sqrt{Nt}}{2\sigma_{\text{obs}}\sqrt{2N\sigma_{\text{split}}^2 + 8\sigma_{\text{split}}^2\sqrt{Nt}}}\right) \leq \Pr(\text{correct}) \leq \Phi\left(\frac{2\epsilon N\sigma_{\text{split}}^2 + 2\sigma_{\text{split}}^2\sqrt{Nt}}{2\sigma_{\text{obs}}\sqrt{2N\sigma_{\text{split}}^2 - 8\sigma_{\text{split}}^2\sqrt{Nt}}}\right) \quad (11)$$

where $t = \frac{\log(6/\delta)}{\min(C_1, C_2)}$.

## B    PROOF OF COROLLARY 1

If we assume $N$ is large, then $N \gg \sqrt{N}$. If we ignore such $\sqrt{N}$ terms, the left and right-hand sides of the inequality in Theorem 1 become the same. In addition to the fact that the Gaussian CDF is smooth, we have

$$p_{success} \approx \Phi \left( \frac{2\epsilon N \sigma_{\text{split}}^2}{2\sigma_{\text{obs}} \sqrt{2N \sigma_{\text{split}}^2}} \right) = \Phi \left( \frac{\epsilon \sqrt{N} \sigma_{\text{split}}}{\sigma_{\text{obs}} \sqrt{2}} \right)$$

The Gaussian distribution is a well-known example of a stable distribution. In other words, the average of Gaussians is Gaussian, and more specifically, we have

$$\sigma_{\text{split}}^2 = \frac{\sigma^2}{m}$$

Plugging this in, we have

$$\Phi \left( \frac{\epsilon \sqrt{N} \sigma_{\text{split}}}{\sigma_{\text{obs}} \sqrt{2}} \right) = \Phi \left( \frac{\epsilon \sigma}{\sigma_{\text{obs}}} \cdot \sqrt{\frac{N}{2m}} \right)$$

as desired.

## C ADDITIONAL EXPERIMENTS

We extend Figure 3 with the models Pythia-70m and Pythia-1b, with substantially similar results, and finding only one failure of our method (PubMed Abstracts, Pythia-70m), corresponding to a success of dataset inference.

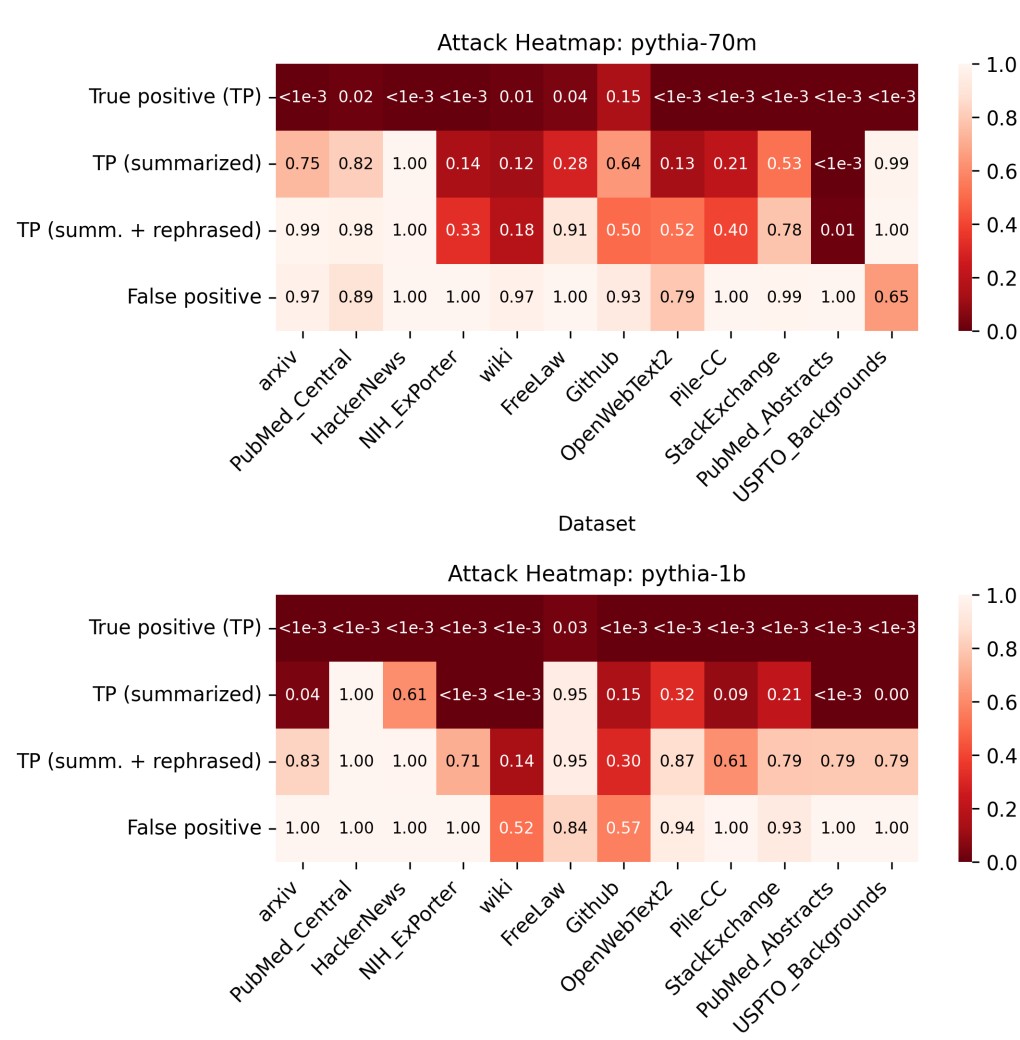

Figure 5: Extensions to Figure 3, with Pythia-70m and Pythia-1b.

## D TOKEN COUNTS FOR SUMMARIZATION AND REPHRASING

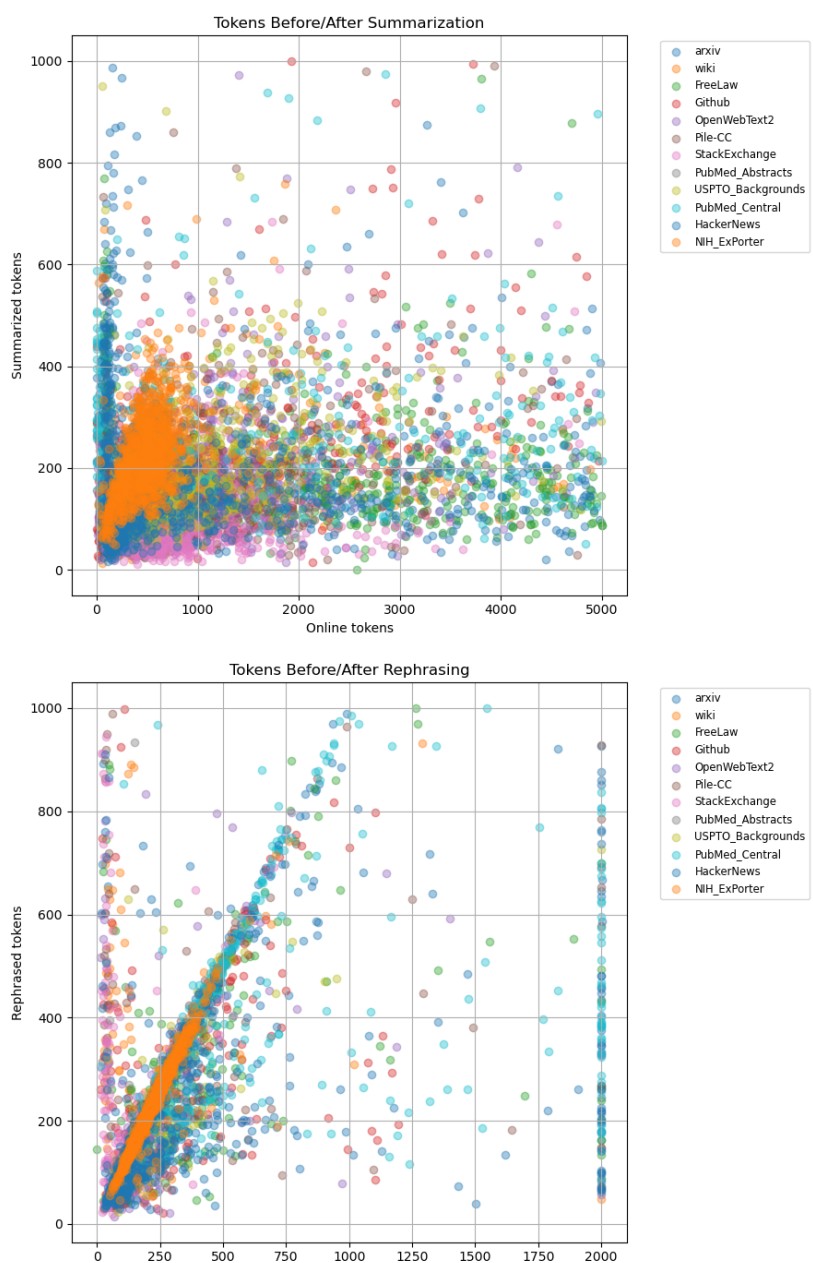

Figure 6: We visualize how the token counts change for our long text operations, both summarization and rephrasing. In the case of summarization, we observe a substantial decrease in token count by a factor of approximately 3. For rephrasing, we observe that the token counts remain about the same, as expected.

We can see in Figure 6 that token counts substantially decrease following summarization. In addition, we allow for a maximum of 2000 new tokens to be generated for each summary, and we found that it was rare for summaries to reach close to the 2000-token limit. For models other than Mistral-7B-Instruct-v0.3, we observed that summaries sometimes repeated themselves or diverged from the original text till they hit the token limit. This repetition did not occur with Mistral-7B-Instruct-v0.3, raising our confidence in a coherent/high-quality summary.

# E  METHOD EXAMPLES

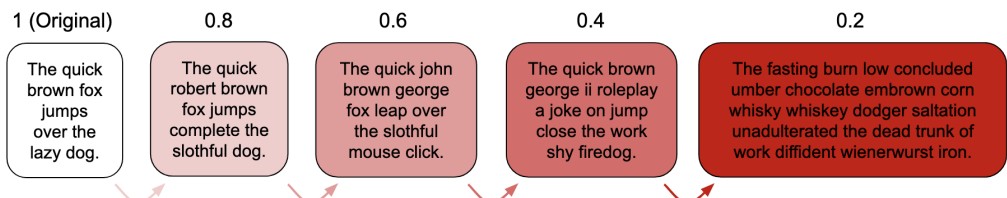

Figure 7: Summarization and rephrase example.

Figure 8: Synonym substitution example.

# F  HYPERPARAMETERS

Table 2: Hyperparameters for our paper.

Pile Experiment

| Hyperparameter | Value |
|---|---|
| Learning rate | $1.5/\theta$ |
| Batch size | 4 |
| Number of epochs | 2 |
| Weight Decay | 0.01 |
| Warmup Steps | 25 |
| Seed | 42 |

Benchmark Experiment

| Hyperparameter | Value |
|---|---|
| Learning rate | 0.0005 |
| Batch size | 4 |
| Maximum Steps | 4000 |
| Weight Decay | 0.01 |
| Warmup Steps | 100 |
| Seed | 42 |

Summarization, Rephrase

| Hyperparameter | Value |
|---|---|
| Temperature | 0.7, 0.8 |
| Top P | 0.9 |
| Max New Tokens | 2000 |
| Seed | 42 |