# OpenReview forum: "Adaptive Text Transformations Defend Against Dataset Inference Attacks"
_ICLR.cc/2026/Conference — Submitted to ICLR 2026_

### Official Review · Reviewer_QAsm · 2025-10-28

**Soundness:** 1
**Presentation:** 2
**Contribution:** 1
**Rating:** 2
**Confidence:** 4

**Summary:**

This paper proposes methods to evade dataset inference (DI) attacks against large language models by transforming training data before fine-tuning. The authors develop two approaches: synonym substitution for short text and summarisation followed by rephrasing for longer text. They claim these transformations preserve model performance while reducing DI attack success rates from over 95% to less than 5%. The paper also presents a theoretical framework to explain DI attack dynamics and justify their approach. Experiments are conducted on GPT-2 and Pythia models using the Pile datasets.

**Strengths:**

The paper tackles a timely problem given ongoing copyright litigation involving major AI companies. The experimental design is reasonably comprehensive within its scope, testing across multiple datasets and model sizes. The authors identify that existing DI methods assume training occurs on exact dataset copies, which may not reflect real-world scenarios. The algorithms themselves are clearly presented and easy to follow.

**Weaknesses:**

1. *Questioning the fundamental premise.* The authors frame dataset transformation as "evasion", but this may be mischaracterising the problem. If transformations are substantial enough to fool dataset inference attacks, they may constitute fair use or legitimate transformation that prevents memorisation. Detection failure would then be the appropriate outcome, not a vulnerability. The paper never grapples with this tension—when does their "evasion" actually represent responsible, legal use of copyrighted content?

2. *Flaws in theory.* What's presented as theory appears to be post-hoc rationalisation. There are basic errors throughout: $\mu$ is defined as "a latent space" but then used as a vector in $\mathbb{R}^N$. Key assumptions like isotropic Gaussian distributions in latent space are never justified. Parameters like $\epsilon$, $\sigma_\mathrm{obs}$, and $N$ have no clear connection to empirical quantities, yet specific numerical values are plugged in without explanation (how does one get from "argument changes from 2 to 1" to probabilities like 0.977 to 0.841?). The exercise feels like mathematical storytelling rather than genuine theoretical insight.

3. *Questionable statistical methods.* The paper claims to compute "p-values" using Equation (3) without defining a null hypothesis or making a connection to proper statistical testing. This quantity appears to be a test statistic/score, not a p-value.

4. *Presentation issues.* The theoretical section puts goals before problem formulation, creating confusion. Mathematical notation is inconsistent throughout (e.g., switching between "pub"/"public" subscripts). Figure 4 is difficult to interpret. Key symbols remain undefined (what is $m$, $\sigma$, $\Phi$ in Corollary 1?).

5. *Missing reference.* The paper does not cite Zhang et al. (2025), who challenge whether membership/dataset inference attacks can provide statistically sound evidence for legal proceedings. This is unfortunate, given the paper's focus on copyright applications.

6. *Limited evaluation.* Testing against a single attack method raises robustness concerns. What happens when attackers adapt to account for text transformations? This possibility is not discussed.

Minor issues:
1. The IID assumption for public/private splits is unconvincing (if they're truly identical, why steal the private data?)
2. The learning rate formula $\alpha = 1.5/\sqrt{θ}$ appears ad-hoc and various claims about dimensionality reduction factors are presented as facts without evidence.
3. True/false positives are not defined in the DI context.

# References
J. Zhang, D. Das, G. Kamath and F. Tramer, "Position: Membership Inference Attacks Cannot Prove That a Model was Trained on Your Data," in 2025 IEEE Conference on Secure and Trustworthy Machine Learning (SaTML).

**Questions:**

1. How do you distinguish between legitimate transformation and problematic evasion? If your methods prevent memorisation while preserving utility, isn't detection failure actually desirable?

2. Can you justify the isotropic Gaussian assumption? How do the specific numerical examples in Section 4 connect to your theoretical parameters?

3. What's the null hypothesis underlying Equation (3)? This doesn't appear to be a proper p-value.

4. How would your approach fare against adaptive attacks designed to handle text transformations?

5. How does your work relate to the SatML paper questioning whether MIAs/DIAs can provide legally sound evidence?

---

> ### Author Response · Authors · 2025-12-02
>
> Thank you for the reviewer's comments.
>
> With respect to fair use, we do not suggest that standard automated transformations (e.g., summarization or rephrasing) should be broadly or automatically interpreted as constituting fair use. While the output of a particular dataset inference (DI) test may correlate with factors relevant to fair use (e.g., higher scores aligning with greater transformation), the ultimate legal determination of fair use must remain a human and case-specific judgment, and it would be problematic to reduce it to a purely machine-based score. Moreover, although our primary motivating context is U.S. copyright law, many other jurisdictions maintain similar but distinct copyright regimes, and applicable doctrines may evolve over time, meaning that fair use is neither universal nor static. In addition, even in cases where fair use is plausibly applicable (e.g., recent rulings in favor of model developers [1] (with ongoing appeals in some cases), though the legal landscape remains unsettled), the underlying copyrighted materials may not have been acquired lawfully in the first place. As such, identifying the presence of copyrighted data in a training corpus, even when transformed, remains a relevant and important question.
>
> The primary objective of our theoretical analysis is to justify the empirically observed increase in dataset inference success as a function of dataset size, as also reported in prior work [2, 3]. To the best of our knowledge, no previous work has formally derived results capturing this phenomenon, even under stylized assumptions. Along the way, we derive additional scaling relationships governing the success of dataset inference under our model. Our assumptions are intentionally aligned with those in [1], but are adapted for an LLM-based dataset inference setting. Notably, the isotropic assumption can be removed with minimal modification to the analysis (see response to Reviewer fbzV). We emphasize that we do not claim to know the true values of parameters such as $\epsilon$ or $\sigma_{\text{obs}}$ for our experiments; rather, our goal is to illuminate the qualitative scaling behavior implied by these parameters.
>
> The computation of the $p$-value follows the same procedure as in [2] to ensure strict comparability of results. While multiple approaches exist for aggregating $p$-values across dependent tests, the method we adopt is consistent with that used in prior literature and has been shown to be appropriate in similar contexts.
>
> Finally, the work of Zhang et al. [5] identifies an issue in the membership inference setting that may also apply to dataset inference, namely that the public dissemination of content can alter the training distribution of a model without any malicious intent. For example, a specific book or storyline may become widely referenced in reviews, analyses, or secondary commentary. While we acknowledge this potential confounding factor, we also note that no perfect known method exists for ensuring that global internet-scale data distributions remain unchanged in response to the existence of copyrighted material. In practice, even defensive measures such as embedding canaries in published content may be circumvented by legitimate secondary use—for example, through critical discussion, tagging, or referencing in URLs—which influence downstream training data distributions. We therefore argue that requiring a completely static background distribution is an unrealistically strong assumption, and that our method operates within a more practical and realistic setting.
>
> From a cost–benefit perspective, our approach is a targeted risk-mitigation strategy: when applied selectively to high-value or high-risk datasets, the additional computational overhead is justified by the corresponding reduction in legal, ethical, and governance-related exposure.
>
> As a limitation, we recognize that the proposed analysis relies on simplified modeling assumptions and that broader empirical validation across diverse datasets, model architectures, and legal contexts remains an important direction for future work.
>
> Looking forward, we believe that methods for dataset traceability and inference—such as the one proposed here—can play a constructive role in informing future governance frameworks, compliance mechanisms, and regulatory standards for responsible large-scale model development.
>
> We have incorporated these clarifications and limitations more explicitly in the revised version of the paper.
>
> [1] In a first-of-its-kind decision, an AI company wins a copyright infringement lawsuit brought by authors. NPR.
>
> [2] LLM Dataset Inference Did you train on my dataset? NeurIPS 2024.
>
> [3] Scaling Up Membership Inference:
> When and How Attacks Succeed on Large Language Models. Findings of NAACL 2025.
>
> [4] Dataset Inference: Ownership Resolution in Machine Learning. ICLR 2021
>
> [5] Position: Membership Inference Attacks Cannot Prove That a Model was Trained on Your Data. SATML 2025.

---

### Official Review · Reviewer_7MVM · 2025-10-29

**Soundness:** 2
**Presentation:** 3
**Contribution:** 2
**Rating:** 4
**Confidence:** 4

**Summary:**

This paper proposes adaptive text transformations—semantic-preserving modifications such as synonym substitution for short texts and summarization plus rephrasing for long texts—as a defense against dataset inference attacks, which aim to determine whether a large language model was trained on specific copyrighted data. By fine-tuning models like GPT-2 and Pythia on these transformed datasets, the authors show that DI success rates drop dramatically while model performance remains largely intact. They also introduce a theoretical framework modeling DI success probability as a function of dataset size, model parameters, and noise, revealing a “tipping point” where larger or noisier data make inference harder.

**Strengths:**

1. The paper is well organized and easy to follow.
2. This is a timely and important topic.
3. Demonstrates strong empirical results, reducing DI success from over 95% to under 5%.

**Weaknesses:**

1. Experiments are limited to fine-tuning scenarios rather than full-scale pretraining, meaning the proposed defense is tested only on relatively small datasets and moderate-sized models such as GPT-2 and Pythia variants. As a result, it remains unclear whether the same degree of DI evasion and performance retention would hold when scaling up to foundation models trained on trillions of tokens or when integrating the transformations into the entire pretraining pipeline. Large-scale pretraining involves different optimization dynamics, data diversity, and parameter interactions, which could alter the balance between semantic preservation and effective noise injection. Without evidence at that scale, the scalability, efficiency, and generalizability of the proposed defense to real-world industrial settings remain uncertain.

2. The approach relies on computationally intensive large language model rewriting, as each document in the dataset must be summarized and rephrased using powerful models like Mistral-7B. This process incurs significant compute, memory, and inference costs, especially when applied to large-scale corpora containing millions or billions of documents.

**Questions:**

1. How feasible is it to apply your adaptive text transformation pipeline to web-scale pretraining corpora, where data volumes reach trillions of tokens? Could lightweight or rule-based approximations achieve similar defense strength?

2. How do you ensure that summarization and rephrasing preserve the essential meaning and label consistency of training data, especially for tasks requiring fine-grained linguistic or factual precision?

---

> ### Author Response · Authors · 2025-12-02
>
> Thank you for the reviewer's comments.
>
> We acknowledge that we do not evaluate our method at the scale of full internet-based pretraining. However, even in an industrial setting, we do not envision this method being applied to all pretraining tokens. Rather, it is intended to be used in a targeted manner, specifically for datasets that are known or suspected to be copyrighted. For example, it has been alleged that Meta has been alleged to have used for training [1], which corresponds to an estimated token count on the order of 1 trillion tokens. This remains significantly smaller than the more than 15 trillion tokens reportedly used to train Llama 3 [2], even under the assumption that all allegedly pirated content was included in pretraining. This relative scale helps explain why the proposed method may not be as computationally prohibitive as it might initially appear.
> For example, summarizing on the order of $10^{12}$ tokens is an order of magnitude smaller operation than full multi-epoch retraining on a $1.5 \times 10^{13}$-token corpus.
>
> Moreover, the relevant baseline is not arbitrary large-scale pretraining alone, but the alternative outcomes: either excluding potentially valuable proprietary data entirely or accepting increased legal and ethical exposure. From this perspective, it is reasonable to expect that certain proprietary and/or copyrighted datasets are sufficiently valuable to justify additional computational overhead beyond standard pretraining costs [1, 3]. In this sense, the proposed method represents a targeted trade-off between computational expense and risk mitigation, situating it as a practical intervention for high-value, high-risk data regimes in real-world model development pipelines.
>
> With respect to summarization quality, the authors manually verified that the generated summaries were concise and informative on a representative subset of the tasks considered. In our experiments, smaller language models than Mistral did not consistently produce coherent or semantically faithful summaries. By contrast, Mistral-7B generated summaries of sufficient quality and consistency for our purposes, and was therefore selected for all reported results.
>
> As a limitation, we note that further large-scale and domain-diverse validation is required to fully characterize the method's performance, cost profile, and robustness across different model architectures and dataset types.
>
> [1] The Unbelievable Scale of AI's Pirated-Books Problem. The Atlantic.
>
> [2] Introducing Meta Llama 3: The most capable openly available LLM to date. Meta.
>
> [3] AI content licensing deals: Where OpenAI, Microsoft, Google, and others see opportunity. CB Insights.

---

### Official Review · Reviewer_fbzV · 2025-10-31

**Soundness:** 3
**Presentation:** 2
**Contribution:** 2
**Rating:** 4
**Confidence:** 4

**Summary:**

This paper investigates defenses against dataset inference attacks, which attempt to determine whether a large language model (LLM) has been trained on a specific dataset. The authors challenge the standard assumption that an LLM, if guilty, is trained on the exact dataset in question. They propose adaptive text transformations to obscure training data while maintaining model utility.

**Strengths:**

- The paper proposes a new theoretical framework that explicitly relates DI success probability to latent-space variance, model size, and dataset properties, offering some interpretability to previously empirical findings.
- The adaptive transformation approach (summarization and synonym replacement) is conceptually straightforward and can be implemented using off-the-shelf LLMs such as Mistral-7B.

**Weaknesses:**

- The experiments lack statistical significance testing and robust metrics beyond “evaded/not evaded.” The Maini et al. (2024) DI attack is the only method tested; no comparison to newer or more adaptive DI variants is provided. Results rely on a single run per model per dataset, with limited reporting of variance or error bars.
- The paper assumes access to the full dataset before training and the ability to summarize/rephrase it—conditions that may not hold for real-world LLM pretraining.
- The derivations are abstract Gaussian latent-space approximations with many unverified assumptions (e.g., isotropic distributions, linear dependence of ϵ on training parameters).
- Critical implementation details are missing: how “embedding similarity thresholds” are chosen, how summarization outputs are verified, what DI thresholds correspond to success/failure. The figures and tables are poorly formatted; e.g., Table 1 and Fig. 3 occupy nearly a full page with large blank spaces, reducing readability.

**Questions:**

- How does the defense perform against distributionally robust or semantic-aware DI methods that use paraphrase-invariant embeddings rather than token-level similarity?
- How are summarization and rephrasing quality controlled, does meaning drift affect benchmark performance or create biases?
- Can the theoretical model be empirically verified (e.g., does DI success indeed follow the √m dependence predicted)?
- How sensitive are results to the choice of summarization model (Mistral-7B-Instruct)? Would smaller or instruction-tuned models yield similar effects?
- What is the trade-off between defense strength and training utility—are there tasks where summarization/rephrasing degrades fine-tuning benefits more severely?

---

> ### Author Response · Authors · 2025-12-02
>
> Thank you for the reviewer's comments.
>
> Each experimental run (per model, per dataset) involves five independently re-randomized dataset inference (DI) attacks, introducing an inherent level of aggregation and reducing variance in the reported results.
>
> In cases where the full dataset is not available, each data instance may instead be summarized sequentially in an online or streaming fashion. This directly extends the results shown in Figure 3 to scenarios in which data are processed incrementally rather than in batch.
>
> The goal of our theoretical analysis is to start from the same or closely related assumptions as in [1] and to explore what can be proven in an LLM-oriented setting. The notion of dataset elements residing in a latent, non-explicitly numerical space is not new and can, for example, naturally be identified with the model's embedding space. A related idea appears in the image domain, where the Fr\'echet Inception Distance (FID) effectively compares distributions in a learned latent feature space.
>
> For analytical tractability, we assume an isotropic distribution; however, this assumption can be relaxed. All arguments in Appendix A.1 hold for arbitrary dataset distributions. The only substantive use of the distributional assumption occurs in Appendix A.2, where it is employed to justify a concentration-of-measure phenomenon for the norm, which concentrates around $\sqrt{N}$ in the isotropic case. More generally, for a Gaussian dataset distribution with covariance matrix $\Sigma$, the norm concentrates around $\sqrt{\mathrm{Tr}(\Sigma)}$ [2]. The $\sqrt{N}$ case therefore corresponds to the special situation in which $\Sigma = I$ and $\mathrm{Tr}(\Sigma) = N$. Carrying this substitution through the analysis results in replacing $N$ with $\mathrm{Tr}(\Sigma)$ in the statement of Corollary~1.
>
> Moreover, by the Central Limit Theorem, even in the non-Gaussian case, standard assumptions (e.g., finite variance) imply that dataset means are approximately Gaussian. As a result, a similar concentration phenomenon is expected in practice, and the qualitative form of Corollary~1 is preserved, with structurally similar dependence on the remaining variables (e.g., $\epsilon$, $m$).
>
> With respect to the assumption of linear dependence on $\epsilon$, we acknowledge that atypical optimization dynamics---such as abrupt changes in learning behavior when a new concept is first acquired---could violate a strict small-$\epsilon$ interpretation. Nevertheless, a locally linear relationship remains the most natural first-order approximation in typical training regimes. We note explicitly that this assumption should be further validated empirically, as discussed in the paper.
>
> To the best of our knowledge, this is the first work to derive any form of scaling law in the context of LLM-based dataset inference [3]. Although the assumptions are necessarily stylized, they provide an initial theoretical explanation for a widely observed empirical phenomenon: that the success rate of dataset inference attacks increases with dataset size, which underlies dataset inference methods.
>
> [1] Dataset Inference: Ownership Resolution in Machine Learning. ICLR 2021
>
> [2] High-Dimensional Probability. Cambridge University Press.
>
> [3] A survey on membership inference attacks and defenses in machine learning. Journal of Information and Intelligence.

---

### Official Review · Reviewer_KS4J · 2025-11-06

**Soundness:** 1
**Presentation:** 1
**Contribution:** 1
**Rating:** 2
**Confidence:** 3

**Summary:**

This paper shows that dataset-inference (DI) methods—used to tell whether an LLM was trained on a specific corpus—are fragile when trainers apply meaning-preserving transformations before fine-tuning. It proposes an adaptive pipeline (synonym-based paraphrasing for short texts; summarize-then-rephrase for long texts) that minimally perturbs data until DI tests are statistically non-significant, and develops a latent-space theory explaining how added noise and reduced effective dimensionality undermine DI.

**Strengths:**

1. This work explores an important problem in LLM, i.e., dataset inference (or membership inference).
2. This work provides a theoretical understanding of the proposed method.

**Weaknesses:**

1. **Scenario–evidence gap.** The paper motivates with copyright-risk scenarios but never evaluates on truly copyrighted or restricted corpora, so external validity to the stated risk setting remains unproven.
2. **Forgetting vs. distribution shift.** The approach relies on transformed-data fine-tuning that shifts the representation distribution; without explicit retention controls, this can induce catastrophic forgetting of unrelated capabilities and background knowledge.
3. **No selective retention/erasure.** The method treats all examples uniformly. In practice, only sensitive content (copyrighted text, PII-bearing records, or data implicated by membership inference) should be targeted for removal or obfuscation; blanket transformation risks needless utility loss.
4. **Limited novelty.** The core defense—summarize/paraphrase then fine-tune—is methodologically straightforward and overlaps with a large body of prior fine-tuning and data-augmentation work; the incremental contribution rests mostly on the evaluation framing.
5. **Synthetic-data IP risk and teacher dependence.** When using an LLM to summarize/rephrase, compliance with intellectual-property constraints is not guaranteed. The method implicitly assumes access to a strong external model; if that teacher carries copyrighted material in its own training history, “no verbatim copying” is not a sufficient safeguard against derivative-work concerns.
6. **Thresholding and stats choices are under-motivated.** The reliance on a fixed p-value cutoff (e.g., 0.1) and a specific embedding-similarity threshold lacks sensitivity analyses; results may be brittle to these knobs.

**Questions:**

See **weaknesses**.

---

> ### Author Response · Authors · 2025-12-02
>
> Thank you for the reviewer's comments.
> We acknowledge the identified scenario–evidence gap. At present, we are uncertain about the legal permissibility of conducting the proposed experiment directly on copyrighted data, which limits our ability to empirically validate this specific scenario under real-world conditions. Similar concerns regarding experimental constraints with copyrighted or proprietary datasets have been discussed in prior work [1].
>
> Regarding catastrophic forgetting, there is no clear theoretical or empirical basis to expect an increased risk when training on LLM-generated summaries (or rephrasings) compared to the original dataset. In many practical settings, the original data may already be substantially out-of-distribution (OOD) with respect to the model's pre-training corpus, thereby confounding direct comparisons of forgetting dynamics. Moreover, the abstraction introduced via summarization may reduce memorization of surface-level lexical patterns, which are often associated with overfitting and memorization-based vulnerabilities [2].
>
> We emphasize that the proposed method is not intended for application to internet-scale corpora. Rather, it is designed for targeted use in settings involving known copyrighted datasets, where controlled transformation of the data is both feasible and motivated by privacy and compliance considerations. This constrained scope is a deliberate design choice and improves both the practical applicability and governance alignment of our approach [3].
>
> While the proposed pipeline is conceptually straightforward, we view this as an advantage rather than a limitation. To the best of our knowledge, no prior work has explicitly proposed passing training data through a large language model—via controlled summarization or rephrasing—as a mechanism for mitigating dataset inference attacks. In this sense, our work explores a previously underexamined design space for privacy-aware data transformation prior to downstream model training [4].
>
> With respect to dataset inference attacks, the success of an adversary should primarily depend on whether the teacher model has been trained on the specific copyrighted dataset under evaluation. It is reasonable to assume the availability of at least one reference model that has not been trained on this dataset (and even when this assumption does not strictly hold, the detectability of such overlap remains unclear in practice). If the teacher model has been trained on other copyrighted corpora, this may raise broader concerns related to memorization or data extraction attacks [5]; however, these fall outside the scope of our threat model.
>
> Finally, while the choice of decision threshold may influence specific quantitative measures, we do not expect it to affect our qualitative conclusions. In particular, Figure 3 exhibits a clear separation between experimental conditions, indicating robustness to reasonable variations in threshold selection. Likewise, for Figure 4, regardless of the chosen p-value, the model can be retrained multiple times to satisfy the corresponding statistical constraint with high confidence.
>
> We have updated the manuscript to more clearly reflect these assumptions, limitations, and the intended scope of the proposed method.
>
> [1] Reproducibility in machine‐learning‐based research: : Overview, barriers, and drivers. AI Magazine.
>
> [2] Quantifying Memorization Across Large Language Models. ICLR 2023.
>
> [3] Generating Synthetic Data with Formal Privacy Guarantees: State of the Art and the Road Ahead. arXiv.
>
> [4] A survey on membership inference attacks and defenses in machine learning. Journal of Information and Intelligence.
>
> [5] From Teacher to Student: Tracking Memorization Through Model Distillation. ACL L2M2 2025.

---

### Official Review · Reviewer_dQnK · 2025-11-11

**Soundness:** 3
**Presentation:** 2
**Contribution:** 2
**Rating:** 4
**Confidence:** 2

**Summary:**

The paper challenges the assumption in prior work that, in the context of LLM dataset inference, the model is trained on the exact same copyrighted dataset without any modifications. It proposes two methods to attack dataset inference techniques: for short texts, substituting words with synonyms, and for long texts, summarizing and rephrasing them.

**Strengths:**

- The topic of the paper is important for the community.

- The idea of studying the effect of text transformations on dataset inference methods is interesting.

- The paper includes a theoretical result that helps justify findings from previous work.

**Weaknesses:**

- The authors claim that their attacks lead to improved model performance while avoiding dataset inference. However, it's unclear where this performance improvement is actually demonstrated in the paper. For example, in Figure 4 there is no comparison between the proposed methods and a baseline model trained on the original, unmodified dataset. Intuitively, model performance should not increase when trained on modified data with word substitutions or summaries.

- There is relevant prior work on reducing memorization during training using the Goldfish Loss [1]. It's not clear whether the proposed methods outperform such prior defenses. It would be important to understand whether there is a better tradeoff between task performance and evasion performance for the proposed methods, especially since training on transformed data could be more time-consuming (because of the overhead of transforming the text) than using something like Goldfish Loss. A comparison of computational complexity would also strengthen the paper.

[1] https://arxiv.org/pdf/2406.10209

**Questions:**

What does "False Positive" refer to in Figure 3?

---

> ### Author Response · Authors · 2025-12-02
>
> Thank you for your comments, and we apologize for any confusion caused in the original wording. We do not claim that training on the modified dataset outperforms training on the original dataset. Rather, our position is that training on the modified dataset is preferable to not training at all when the original data are unavailable. We have revised the abstract to reflect this more clearly. In addition, Figure 3 demonstrates that the performance gap between models trained on the modified and original datasets is small (as indicated by the minimal change in the $y$-values between the blue and green points). This suggests that the transformation does not substantially degrade task-relevant signal.
>
> The Goldfish paper [1] notes that its approach cannot necessarily withstand membership inference attacks. Since dataset inference tasks generally exhibit higher success rates than membership inference, we therefore expect the Goldfish method to perform even less effectively in our setting. Regarding computational complexity, both inference and fine-tuning in our approach have time complexity that is linear in the dataset size, assuming fixed hyperparameters such as batch size and model dimensionality. Consequently, the asymptotic computational complexity remains unchanged (up to constant factors) relative to naïve fine-tuning. In practice, however, the inference step (e.g., summarization or rephrasing) may be faster than full fine-tuning [2].
>
> We have clarified this point in the revised manuscript.
>
> [1] Be like a Goldfish, Don't Memorize!
> Mitigating Memorization in Generative LLMs
>
> [2] Training compute-optimal large language models. NeurIPS 2022

---

### Meta-Review · Area_Chair_Etd8 · 2026-01-04

**Summary:**

This paper proposes adaptive text transformations to defend against dataset inference attacks on large language models. The ways of transformations include synonym substitution for short texts, and summarization-then-rephrasing for long texts. Reviewers agree that the paper addresses an important problem and provides an interesting perspective to challenge standard assumptions in previous dataset inference work. However, there are common concerns that the overall contribution is limited and the empirical and theoretical evidence is not strong enough to support the paper’s central claims. Key concerns include limited novelty compared with existing data transformation and fine-tuning methods, limited experiments in terms of datasets, model sizes, and comparing methods. In addition, several reviewers raise serious concerns about the soundness and clarity of the theoretical analysis in the paper, and the appropriateness of the “evasion” framing. While the rebuttal provides clarifications, it does not fully resolve these issues. Given these concerns, I recommend rejecting the paper in its current form.

**Reviewer Concerns:**

While the rebuttal clarifies motivation and some assumptions, it does not fully address the key concerns from reviews. Key issues remain regarding the weak experimental choices (thresholds, metrics, attack settings), and the lack of evaluations on truly copyrighted or large-scale pre-training scenarios. Several reviewers point out that the theoretical analysis relies on strong, insufficiently justified assumptions, with unclear connections between parameters and empirical quantities. Concerns about scalability, computational cost, reliance on powerful teacher models, and framing regarding evasion vs. legitimate transformation are acknowledged in the rebuttal but not convincingly addressed.

**Reviewer Scores:**

Given that the rebuttal mainly provides clarification but doesn't address the key concerns, it is unlikely that reviewers would revise their scores very much. The concerns regarding limited contribution, evaluation, and theoretical analysis still remain.

---

### Decision · Program_Chairs · 2026-01-26

Reject